# Tailoring oral targeted therapies dosage in lung cancer: A systematic review of pharmacokinetics studies on renal and hepatic impairment

Harri Hardi[1], Zahra Fitrianti[1], Karen Elliora Utama[2], Ananda Pipphali Vidya[2],
Nurul Gusti Khatimah[3], Kevin Aristyo[4], Hana Khairina Putri Faisal[5], Vivian Soetikno[6]*

1 Clinical Pharmacology Specialist Study Program, Faculty of Medicine, Universitas Indonesia, Jakarta, Indonesia, 2 Faculty of Medicine, Universitas Indonesia, Jakarta, Indonesia, 3 Doctoral Program in Biomedical Sciences, Faculty of Medicine, Universitas Indonesia, Jakarta, Indonesia, 4 International Ph.D. Program in Cell Therapy and Regenerative Medicine, College of Medicine, Taipei Medical University, Taipei City, Taiwan, 5 Department of Pulmonology and Respiratory Medicine, Faculty of Medicine, Universitas Indonesia, Jakarta, Indonesia, 6 Department of Pharmacology and Therapeutics, Faculty of Medicine, Universitas Indonesia, Jakarta, Indonesia

* petravivian09@gmail.com, Vivian.soetikno@ui.ac.id

## Abstract

### Background

Lung cancer is the leading cause of cancer-related deaths worldwide, and stage IV lung cancer is frequently managed with targeted therapy. Renal and hepatic impairment frequently coexist with cancer, often requiring a reduction in targeted therapy dosage. This systematic review assesses the appropriateness of current targeted therapy dosage adjustments in individuals with hepatic and renal impairment by comparing package insert recommendations with available pharmacokinetic studies.

### Methods

We reviewed the most recent guidelines from the National Comprehensive Cancer Network (NCCN) on the use of non-monoclonal antibody targeted therapy. We also examined all package inserts for information on dose adjustment in cases of hepatic and renal impairment. We then systematically searched for studies that involved pharmacokinetic analysis in populations with hepatic or renal impairment, as well as those undergoing hemodialysis and peritoneal dialysis.

### Results

We identified 44 studies from 21 oral lung cancer therapies that met the inclusion criteria. We developed 13 new recommendations and updated 7 existing ones regarding targeted therapy dose adjustment in cases of hepatic and renal impairment compared to the information provided in the package insert. Several drugs have not published their pharmacokinetic results in a scientific journal, which limits access to

**Data availability statement:** All relevant data are within the paper and its Supporting information files.

**Funding:** The author(s) received no specific funding for this work.

**Competing interests:** The authors have declared that no competing interests exist.

their appropriateness. Moreover, there is a lack of research on pharmacokinetic analysis of targeted therapy in patients undergoing hemodialysis and peritoneal dialysis.

## Conclusions

Adjusting the dosage of targeted therapy in hepatic and renal impairment based on pharmacokinetic analysis is essential to broaden the usage, improve effectiveness, and minimize side effects. Further pharmacokinetic research on the usage in unstudied populations is strongly advised.

## Prospero registration number

CRD42024518123.

## Introduction

Lung cancer is the primary cause of cancer-related deaths globally, with over 2.0 million new cases and around 1.8 million deaths in 2020 [1]. For patients with stage IV lung cancer, lung cancer cells will be tested for specific mutations, and targeted therapy will be used if any genes are found to be mutated [2,3]. Among patients with stage IV lung cancer, 3.2% of patients have mild liver disease and 5.7% have renal disease [4]. Physiological changes in liver and renal disease can alter drug metabolism and elimination, necessitating dose reduction of drugs, including targeted therapy.

In addition to requiring dose reduction in cases of liver impairment, lung cancer frequently spreads to the liver. 17.5% of small cell lung cancer (SCLC) patients and 3.8% of non-small cell lung cancer (NSCLC) patients have liver metastases when diagnosed [5]. In liver metastasis, cytochrome P450 levels were reduced by up to 6.6 times compared to healthy controls, leading to a decreased ability to metabolize targeted therapy [6]. Furthermore, targeted therapy often results in hepatotoxicity as a side effect. For instance, 5,5% of tyrosine kinase inhibitor (TKI) users will develop high-grade hepatotoxicity [7]. This hepatotoxicity occurrence also exacerbates liver function impairment.

Regarding renal impairment, the median overall survival (OS) for lung cancer patients was 11.1 months for stage 3 chronic kidney disease (CKD), 6.0 months for stage 4 CKD, and 4.7 months for stage 5 CKD [8]. Adjusting drug dosages for patients with CKD may impact OS rates. Systematic reviews indicated that a range of 1% to 43% of drugs were not adjusted correctly for renal impairment in outpatient settings and 13.7% in inpatient settings [9,10]. Dose adjustment for drugs in cases of hepatic and renal impairment is usually detailed in their package insert. However, targeted therapy is a novel drug and pharmacokinetic research that supports reducing drug dosage in cases of hepatic and renal impairment are still limited, with the most recent recommendation review published in 2019 [11]. To our knowledge, this is the first systematic review to evaluate targeted therapy pharmacokinetic studies in hepatic and kidney impairment.

This systematic review of pharmacokinetic studies aims to enhance the precision of treatment protocols and improve patient outcomes in vulnerable populations, particularly those with renal and liver failure. Our focus is to assess whether the existing dosages for targeted therapy, as indicated on the package insert, are appropriate based on available pharmacokinetic studies. Additionally, this research is intended to facilitate clinicians in establishing guidelines within their hospitals regarding the use of these medications in patients with liver or kidney disorders.

## Methods

The systematic review followed the Preferred Reporting Items for Systematic Reviews and Meta-Analyses (PRISMA) guidelines [12]. The adherence of this study to the PRISMA guideline can be seen in S1 File. The study protocol can be found on The International Prospective Register of Systematic Reviews (PROSPERO) database with the registration number CRD42024518123.

### Initial data extraction

Prior to initiating the systematic review, we identified targeted therapies indicated for lung cancer according to National Comprehensive Cancer Network (NCCN) latest guidelines for small cell and non-small cell lung carcinoma [2,13]. We followed with the current Food and Drug Administration (FDA) status of these drugs, excluding those that have not been utilized for lung cancer due to efficacy or safety concerns. We excluded immunotherapy and antibody monoclonals used in lung cancer targeted therapy because they are metabolized and eliminated without involving the liver and kidneys [14], making a systematic review of their pharmacokinetic properties unnecessary. We subsequently investigated the dosage of all drugs, including dosages for renal dysfunction, hepatic dysfunction, hemodialysis (HD) and/or continuous ambulatory peritoneal dialysis (CAPD), and considerations for hepatotoxicity from each drug package insert (https://dailymed.nlm.nih.gov/dailymed/).

### Study eligibility criteria

This systematic review included four main types of studies: (1) Physiologically based pharmacokinetic (PBPK) modelling, which uses mathematical prediction to predict the drug's pharmacokinetics based on physiological variables [15]. This type of study is essential for generating predictive insights into how hepatic and renal impairment affect drug behavior, (2) Population pharmacokinetics (PopPK) studies, these studies involve analyzing data from all individuals in a population (based on clinical trial and cohort studies) simultaneously using a nonlinear mixed-effect model to study pharmacokinetics at the population level [16], (3) pharmacokinetic (PK) studies, these studies are essential for analyzing drug levels in the blood over time using data from numerous patients with renal or hepatic impairment, comparing them to individuals with normal function. This method aids in comprehending the differences and trends in drug metabolism and excretion among this group of individuals, (4) Pharmacokinetic case reports or case series in hepatic or renal impairment, these studies provide valuable insights into the pharmacokinetic of targeted therapies in patients with renal or hepatic impairment, focusing on metabolism and elimination processes.

Studies evaluating renal impairment were included if they used the estimated glomerular filtration rate (eGFR) calculated by the CKD-EPI. Studies evaluating liver impairment were included if they utilized the Child-Pugh score, which ranges from mild (CP A) to severe (CP C). Other methods to assess renal and liver impairment were also considered to be included, such as National Cancer Institute classification (NCIc). This study also investigated the use of targeted therapy in patients undergoing HD or CAPD. The primary objective of evaluating HD or CAPD is to determine the elimination in these modalities.

The primary outcome is the area under the curve (AUC) from time zero to infinity ($AUC_{0-\infty}$). This pharmacokinetic parameter was chosen because it offers a thorough assessment of the drug's overall presence in the body throughout a period of time. We also evaluated the maximum concentration ($C_{max}$) of the drug in the bloodstream, a critical factor associated with higher toxicity risks. Given that oral targeted therapy is administered in multiple doses, we considered a study

to be more reliable when it evaluated pharmacokinetic parameters at various time intervals, particularly if the attainment of steady state for these drugs is estimated. We also evaluated the effectiveness and safety of these medications in individuals with hepatic and liver dysfunction. Studies were excluded if not assessing pharmacokinetic parameters.

## Search strategy and study selection

We searched articles systematically from inception until May 1, 2024, through PubMed, Scopus, and Web of Science based on PECO "patient receiving targeted therapy that is used in lung malignancy", "patient with liver impairment, renal impairment, HD, or CAPD", and "pharmacokinetic parameters". No language restriction was applied to this systematic review. Comprehensive keyword usage is provided in Table 1.

The results of our systematic search were gathered and organized using Mendeley Desktop (Glyph & Cog LLC, 2020). The software automatically eliminated duplicate studies. Subsequently, two independent authors (HH and ZF) manually removed any duplicated studies. Two authors (HH and ZF) independently evaluated all titles and abstracts based on inclusion and exclusion criteria; any disagreements were resolved through discussion with a third author (VS). We recorded all reasons for excluding articles. We acquired the complete texts of the studies that met our criteria through a search or by contacting the corresponding author for full text availability. In the case of missing data, we reached out to the corresponding author to inquire about data availability. The full-text screening method was conducted in the same manner as the title and abstract screening method.

## Data extraction and quality assessment

We systematically selected suitable articles and extracted data from all of them according to standard reporting in pharmacokinetic studies [17]. We extracted data regarding ethical approval, study population, dosing, sampling schedule, analytical method, lower limit of quantification, and pharmacokinetic modelling strategies in the method. We obtained the results in the form of patient demographic data, concentration-time relationships, the process of model development, and the final model along with its evaluation plots. We utilized Web Plot Digitizer (https://apps.automeris.io/wpd/) to extract precise values from graphs that did not have exact data points.

Two authors, HH and ZF, independently extracted the data and any other relevant information pertaining to the results. We reached a consensus with the third author (VS) to resolve the disagreement. The consensus results were entered into a word processor, and another author (NGK) verified all data input. If modifications were made, the other three initial authors were consulted regarding their suitability.

We utilize the Risk of Bias (RoB) tool developed by Murad for case reports and case series [18]. PK studies utilize the NOS (Newcastle–Ottawa Scale) RoB tool for case-control studies [19]. The PROBAST (Prediction model Risk Of Bias ASsessment Tool) RoB tool was used for PopPK and PBPK studies [20]. Two independent authors (KEU and APV) assessed the RoB, and any discrepancies were resolved by discussing with a third author (HKPF). We obtained participant and intervention characteristics, along with these publications' funding sources, as a method of assessing RoB.

## Measures of treatment effect

This systematic review focuses on $AUC_{0-\infty}$ and $C_{max}$ parameters as indicators of treatment efficacy and safety. We also extracted other data that can support pharmacokinetic results, such as half-life and clearance, along with metabolite and unbound drug pharmacokinetic parameters. These results were displayed as a table with a comparison for drug label and pharmacokinetic studies. We chose not to conduct a meta-analysis due to the presence of significant heterogeneities among the studies, including variations in drugs, treatment effects, and study populations. Therefore, meta-analysis is considered inappropriate for the current study design. We presented all the results in a table, comparing them with the currently accepted dosage in renal and hepatic impairment. We utilized BioRender (https://www.biorender.com/) to create a visual representation of the key findings from our systematic review. All methods employed in this manuscript are elucidated in Fig 1.

**Table 1. Proposed initial drug dosage in normal renal/hepatic function, hepatic impairment, and renal impairment based on package insert and pharmacokinetic studies.**

| No | Drug class | Drug name/ PubChem CID | Dosage | Hepatic impairment dosage (LoC) | Renal impairment dosage (LoC) | HD/CAPD dosage (LoC) |
|---|---|---|---|---|---|---|
| 1 | EGFR inhibitor | Afatinib/ 10184653 | 40 mg QD | Mild to moderate: normal dose<br>Severe: has not been studied | 30-89: normal dose<br>15-29: 30 mg QD<br><15: has not been studied | HD: 30 mg QD (low)[a] |
| 2 | | Erlotinib/ 176870 | 150 mg QD | Mild to moderate: normal dose, closely monitor<br>Severe: has not been studied | Normal dose (moderate)[a] | HD: Normal dose (low)[a] |
| 3 | | Dacomitinib/ 11511120 | 45 mg QD | Normal dose | 30-89: normal dose<br><30: has not been studied | Has not been studied |
| 4 | | Osimertinib/ 71496458 | 80 mg QD | Mild to moderate: normal dose<br>Severe: has not been studied | Normal dose (Moderate)[b] | HD: Normal dose (moderate)[a] |
| 5 | | Gefitinib/ 123631 | 250 mg QD | Mild: normal dose (very low)[a]<br>Moderate to severe (non-cirrhosis): normal dose (very low)[b]<br>Moderate to severe (cirrhosis): normal dose, closely monitor (very low)[b] | Normal dose | Normal dose in HD and CAPD (low)[a] |
| 6 | ALK inhibitor | Alectinib/ 49806720 | 600 mg BID | Mild to moderate: normal dose<br>Severe: 450 mg BID | Normal dose (very low)[b] | HD: Normal dose (low)[a] |
| 7 | | Brigatinib/ 68165256 | 90 mg QD for 7 days, then increase to 180 mg QD | Mild to moderate: normal dose<br>Severe: 60 mg QD for 7 days, then increase to 120 mg QD | 30-89: normal dose<br>10-29: 60 mg QD for 7 days, then increase to 90 mg QD (low)[b]<br><10: has not been studied | Has not been studied |
| 8 | | Ceritinib/ 57379345 | 450 mg QD | Mild: normal dose<br>Moderate to severe: has not been studied | 30-89: normal dose (very low)[a]<br><30: has not been studied | Has not been studied |
| 9 | | Crizotinib/ 11626560 | 250 mg BID | Mild: normal dose<br>Moderate: 200 mg BID<br>Severe: 250 mg QD | 30-89: normal dose<br><30: 250 mg QD | Has not been studied |
| 10 | | Lorlatinib/ 71731823 | 100 mg QD | Mild: normal dose<br>Moderate to severe: has not been studied | 30-89: normal dose<br><30: 75 mg QD (low)[a] | Has not been studied |
| 11 | BRAF/ MEK inhibitor | Dabrafenib/ 44462760 | 150 mg BID | Mild: normal dose<br>Moderate to severe: has not been studied | 30-89: normal dose (very low)[a]<br><30: has not been studied | HD: start as low as possible (75 mg QD) (very low)[a] |
| 12 | | Trametinib/ 11707110 | 2 mg QD | Mild: normal dose<br>Moderate: start with 1.5 mg (very low)[a]<br>Severe: start with 1 mg (very low)[a] | 30-89: normal dose (very low)[a]<br><30: has not been studied | HD: start as low as possible (0.5 mg QD) (very low)[a] |
| 13 | | Vemurafenib/ 42611257 | 960 mg BID | Mild to moderate: normal dose[c]<br>Severe: has not been studied | 30-89: normal dose[c]<br><30: has not been studied | Has not been studied |
| 14 | NTRK inhibitor | Larotrectinib/ 46188928 | 100 mg BID | Mild: normal dose<br>Moderate to severe: 50 mg BID | Normal dose[c] | Has not been studied |
| 15 | | Entrectinib/ 25141092 | 600 mg QD | Mild: normal dose[c]<br>Moderate to severe: has not been studied | 30-89: normal dose[c]<br><30: has not been studied | Has not been studied |
| 16 | MET inhibitor | Capmatinib/ 25145656 | 400 mg BID | Normal dose (low)[a] | 30-89: normal dose[c]<br><30: has not been studied | Has not been studied |
| 17 | | Tepotinib/ 25171648 | 450 mg QD | Mild to moderate: normal dose<br>Severe: has not been studied | 40-89: normal dose (low)[b]<br><40: has not been studied (low)[b] | Has not been studied |

*(Continued)*

**Table 1.** (Continued)

| No | Drug class | Drug name/ PubChem CID | Dosage | Hepatic impairment dosage (LoC) | Renal impairment dosage (LoC) | HD/CAPD dosage (LoC) |
|---|---|---|---|---|---|---|
| 18 | RET inhibitor | Cabozantinib/ 25102847 | 60 mg QD | Mild: normal dose<br>Moderate: 40 mg QD<br>Severe: has not been studied | 30-89: normal dose<br><30: has not been studied | Inconclusive |
| 19 | | Selpercatinib/ 134436906 | <50 kg: 120 mg BID<br>≥50 kg: 160 mg BID | Mild to moderate: normal dose[c]<br>Severe: 80 mg BID[c] | 15-89: normal dose[c]<br><15: has not been studied | Has not been studied |
| 20 | | Pralsetinib/ 129073603 | 400 mg QD | Normal dose (low)[b] | 30-89: normal dose[c]<br><30: has not been studied | Has not been studied |
| 21 | KRAS inhibitor | Sotorasib/ 137278711 | 960 mg QD | Mild: normal dose[c]<br>Moderate to severe: has not been studied | 30-89: normal dose[c]<br><30: has not been studied | Has not been studied |

LOC: level of certainty. [a]New recommendation, not stated in package insert. [b]Recommendation is modified from package insert, based on pharmacokinetics studies. Renal impairment is based on estimated glomerular filtration rate (eGFR). HD: hemodialysis. CAPD: continuous ambulatory peritoneal dialysis. CID: Compound Identification. QD: once daily. BID: twice daily. [c]Recommendation indicated in the package insert without corroborating the available literature.

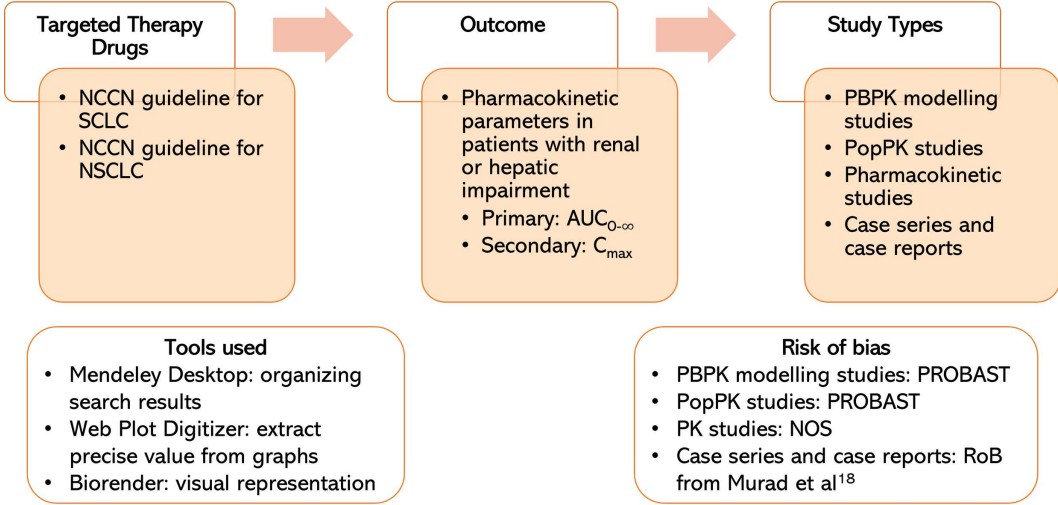

**Fig 1. Graphical visualization of the research methodology.** NCCN = National Comprehensive Cancer Network; SCLC = small cell lung cancer; NSCLC = non-small cell lung cancer; AUC$_{0-\infty}$ = area under the concentration from time 0 extrapolated to infinity; C$_{max}$ = maximum serum concentration; PopPK = population pharmacokinetic; PBPK = physiologically based pharmacokinetic; PROBAST = Prediction model risk of bias assessment tool; NOS = Newcastle–Ottawa Scale; RoB = risk of bias.

We evaluated the level of certainty (LoC) for each new or revised recommendation from the package insert utilizing the GRADE (Grading of Recommendations Assessment, Development and Evaluation) approach. Two independent authors (HH and ZF) evaluated the LoC independently; any disagreements were subsequently deliberated with the third author (VS). We employed GRADEpro GDT (Guideline Development Tool) to create the table concerning LoC from https://gdt.gradepro.org/.

## Results

We identified 502 articles by searching three databases, and the search results for each drug can be found in S2 File. We identified 44 studies that met our PECO criteria after eliminating duplicates and applying inclusion and exclusion criteria to the results. The reasons for excluding records can be seen in Fig 2. We excluded mobocertinib from our search because this drug has been voluntarily withdrawn for EGFR Exon 20 insertion mutation-positive NSCLC due to its ineffectiveness. Furthermore, we identified study characteristics of each article, which are listed in S3 Table in S2 File. We assessed the bias in each study and found that most of them exhibit a low risk of bias. Case reports and case series studies are often biased due to inadequate causality assessment evaluation (3 out of 11) and patient data quality (2 out of 11), which can impact drug pharmacokinetics, such as hypoalbuminemia and concurrent drug use. Potential bias in PK studies is primarily due to a high rate of participants dropping out (8 out of 21) and inadequate sample size per group (5 out of 21). There is a single moderate risk of bias in the PopPK study attributed to the limited sample size. The RoB in case reports and case series was presented in S4 Table in S2 File, RoB in PK studies in S5 Table in S2 File, and RoB in PopPK and PBPK studies in S6 Table in S2 File.

The drug dosing adjustments for hepatic and renal impairment are outlined in S7 Table in S2 File, based on drug labels and studies on impaired renal and hepatic function. In hepatic and renal impairment, we discovered several drug dosing modification techniques, such as lowering the dose and extending the interval. For example, in the general population, crizotinib is administered at 250 mg BID; however, in cases of significant hepatic impairment, this dosage should be lowered to 250 mg QD. Another example of lowering the dosage is the use of trametinib, which is recommended to be reduced to 1 mg QD in cases of significant hepatic impairment from its starting dose of 2 mg QD in the general population. For targeted therapy in liver and hepatic impairment, the maximum initial dosage reduction is 50%. A summary of this information can be found in Table 1.

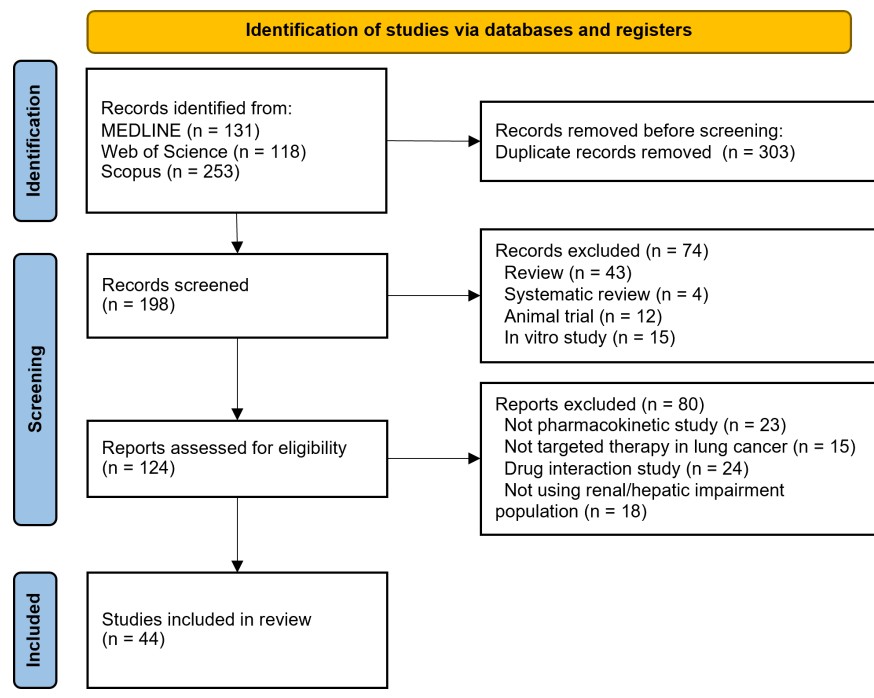

**Fig 2. PRISMA flowchart.**

The elucidation of the LoC for each new or revised recommendations is available in S8, S9, and S10 Tables in S2 File, which cover hepatic impairment, renal impairment, and the HD/CAPD population, respectively. The majority of the LoC is classified as low (9 recommendations), followed by extremely low (7 recommendations), and moderate (3 recommendations). The poor recommendation is due to the absence of assessments utilizing the ROBINS-I (Risk Of Bias In Non-randomised Studies – of Interventions) methodology across all studies. We recognized that this instrument may be unsuitable for our study inclusion due to the pharmacokinetic analysis, which could be challenging if evaluated by ROBINS-I. Moreover, we reduced the certainty if it does not utilize direct PK studies, as modeling could decrease the certainty of the evidence. We detailed the specific considerations for targeted therapy in patients with renal and hepatic impairment, including information on their metabolism and elimination, in the following section.

### EGFR (Epidermal Growth Factor Receptor) inhibitor

About 20% of patients with lung adenocarcinoma have EGFR mutations, which are oncogenic drivers [21]. Erlotinib and Gefitinib, which bind to EGFR reversibly, are the first generation of EFGR inhibitors. Afatinib and dacomitinib, two second-generation EGFR inhibitors, have irreversible binding and may improve progression free survival (PFS) more than first-generation inhibitors, but they also have more adverse effects [22]. Osmertinib, a third-generation EGFR inhibitor, has a higher FPS than first-generation because it can target and activate both EGFR-sensitive and T790M-resistant mutations [23].

**Afatinib.** Afatinib has minimal metabolism, with the unchanged drug mainly excreted in the feces and around 5% excreted in urine [24]. The PopPK studies demonstrated a correlation between elevated alkaline phosphatase (ALP) and aspartate transaminase (AST) levels and an increase in afatinib trough concentrations [25,26]. However, these results were not substantiated in a PK study of afatinib in CP A and B [27]. Hence, the package insert indicated that no modification is required for CP A and B. No study is available in CP C. Using afatinib in CP C patients may exacerbate liver failure, as 9.7% of patients experienced liver test abnormalities and 0.2% of those cases were fatal in regular usage of afatinib. A multidisciplinary team needs to determine the utilization of afatinib in CP C.

The PopPK studies elucidate the contradictory findings regarding the variance in AUC due to renal impairment [25,26]. However, a PK study demonstrated a substantial rise in the AUC in patients with an eGFR of 15–29 compared to the control group (150%, 90% CI: 105–213%) [28]. Given the significant variability of AUC, lowering the initial dosage to 30 mg once daily as advised in the package insert is a sensible decision with more careful monitoring. Renal failure with an eGFR < 15 has not been specifically researched, and other PopPK studies have not exclusively focused on this eGFR range. Therefore, we cannot determine the appropriate dosage for patients with an eGFR less than 15. The package insert for the dialysis patient has not been elucidated. However, case reports and case series studies showed that equivalent trough concentrations were attained with 30 mg of afatinib in NSCLC patients and 40 mg in those with normal renal function [29,30]. Hence, a starting dose of 30 mg is appropriate for initial administration in dialysis.

Given the significant range in AUC, it is prudent to reduce the initial dosage to 30 mg once daily, as recommended in the package insert, accompanied by more diligent monitoring.

**Erlotinib.** Erlotinib is primarily metabolized in the liver by the cytochrome P450 (CYP)3A4 (approximately 80%), CYP1A1, and CYP1A2 and eliminated through feces [31]. High bilirubin and AST levels in patients are associated with reduced erlotinib clearance and prolonged erlotinib half-life [32]. Nevertheless, erlotinib reached its peak concentration earlier in patients with CP B, resulting in a significantly lower $C_{max}$ and no differences in AUC [33]. No reductions are required in CP A, B, or C, according to the package insert. However, our systematic review did not find any studies in CP C, so this statement may be overestimated. It is highly recommended to regularly monitor liver function in patients with pre-existing liver damage due to the elevated risk of hepatotoxicity, as indicated in the package insert. OSI-420, the primary metabolite of erlotinib, also shows no significant difference in CP B [33]. However, a case report demonstrated a notable rise in OSI-420 concentration in individuals with hyperbilirubinemia [34]. This case report analyzed a patient with pancreatic adenocarcinoma, which may result in variations in the metabolism and elimination of erlotinib.

A PK study in renal failure revealed no significant differences in half-life and clearance parameters compared to normal renal function [32]. A PK study in HD patients showed no significant difference in AUC and $C_{max}$, although there was a trend towards a decrease. [35]. The high protein binding affinity of erlotinib (92–95%) may account for the elevated elimination in hypoalbuminemia, a condition frequently seen in HD patients [35]. No recommendation is provided in the package insert regarding renal impairment; however, PK studies demonstrated that the normal dose of erlotinib can be administered in cases of renal impairment and HD [32,35].

**Dacomitinib.** Dacomitinib can be used with a normal dose for patients with liver impairment due to its lack of statistically significant AUC and $C_{max}$ differences in patients with mild to severe liver impairment [36,37]. These phase I studies are limited to patients classified as CYP2D6 extensive or intermediate metabolizers due to dacomitinib being a CYP2D6 substrate. Additional research on individuals classified as CYP2D6 poor metabolizers should be carried out to improve our comprehension across all demographics. No studies have been done on dacomitinib for renal failure, but it is deemed safe in this condition as only 3% of it is excreted in urine. Further research is needed for patients with an eGFR below 30 or undergoing dialysis, as the package insert does not provide recommendations for this group.

**Osimertinib.** Osimertinib and its metabolites are removed from the body through the liver and kidneys, with 68% excreted in feces and 14% in urine [38]. Osimertinib shows reduced exposure in patients with mild and moderate hepatic impairment, with levels of osimertinib and its metabolite being around 30% to 50% lower based on a PK study [39]. The decreased exposure could be a result of reduced bioavailability in hepatic impairment, as shown by the decrease in $C_{max}$, but additional research is needed. While $C_{max}$ and AUC were decreased in the study, a PopPK analysis demonstrated that hepatic impairment did not affect the pharmacokinetics of osimertinib [40]. Osimertinib can be administered with the standard dosage as indicated in the package insert while assessing the clinical effectiveness of osimertinib.

Severe renal impairment (eGFR < 30) caused a 26% increase in AUC. The increase was statistically significant but not clinically relevant due to osimertinib's wide safety margin [41]. A PK study demonstrated a notable decrease in both AUC and $C_{max}$ by 57% and 59%, respectively, when patients with an eGFR < 30 or undergoing HD were administered a reduced dose (40 mg) of osimertinib [38]. Another case report demonstrated that patients undergoing HD can receive a full dose of osimertinib with pharmacokinetic properties similar to those with normal renal function [42]. Another case report demonstrated that administering osimertinib at a daily dose of 80 mg led to a $C_{max}$ level exceedingly twice the standard amount, resulting in severe grade 3 general weakness. Reducing the dose to 240 mg per week decreased this side effect, and administering 480 mg per week did not increase the $C_{max}$ [43]. Although the package insert does not have recommendations for individuals with an eGFR < 15 or undergoing HD, we can consider using a normal dose for this population with careful therapeutic drug monitoring (TDM).

**Gefitinib.** Gefitinib is extensively metabolized in the liver by CYP3A4 and CYP2D6. This leads to an elevation in AUC and a decrease in clearance in patients with cirrhosis following a single dose of gefitinib. Nevertheless, the AUC did not show a statistically significant increase after multiple administrations of gefitinib [44]. Single-dose study involves patients with cirrhosis, while multiple-dose study involves patients with liver metastasis. This population disparity may impact the outcome. The package insert recommends monitoring for adverse reactions in cases of moderate and severe hepatic impairment. The lower dose is not available due to the single dosage preparation of 250 mg.

Gefitinib and its metabolites have a renal excretion rate of less than four percent, indicating their safety in cases of renal failure, as per the package insert. Two studies in HD and one study in CAPD have been conducted [45–47]. They all reported that the $C_{max}$ value was comparable to that of patients with normal renal function. HD removes 11.3% of gefitinib, while the concentration of gefitinib in CAPD fluid is 5.6% [45,47]. Therefore, gefitinib is considered safe in renal impairment.

## ALK (Anaplastic Lymphoma Kinase) inhibitors

About 5% of individuals with NSCLC have an ALK gene rearrangement, which increases the risk of brain metastases. The first ALK inhibitor was crizotinib. When treating brain metastases, second-generation drugs (ceritinib, alectinib, and

PLOS One | https://doi.org/10.1371/journal.pone.0324056   July 29, 2025

9 / 24

brigatinib) show stronger intracranial action compared to crizotinib [22]. Compared to patients treated with crizotinib, those treated with lorlatinib, a third-generation ALK inhibitor, had a significantly longer PFS and a higher likelihood of an intracranial response. However, lorlatinib also has a higher incidence of grade 3 or 4 adverse events due to the altered lipid level [48].

**Alectinib.** Alectinib is metabolized by the enzyme CYP3A4 into its primary active metabolite, M4. The metabolism of alectinib is influenced by variations in the CYP3A4 gene, as demonstrated in an in vivo study [49]. Moderate and severe hepatic impairment resulted in a significant increase in unbound alectinib AUC by 86% and 185%, respectively, compared to the control group. It also led to a significant increase in total alectinib levels but did not significantly affect the $C_{max}$. The pharmacokinetic parameter of M4 was not statistically significant in patients with CP B and CP C [50]. An additional PBPK study confirmed this finding [51]. We recommend increasing monitoring for patients with CP B who are using 600 mg alectinib due to the increased AUC parameters, despite the package insert suggesting only reducing the dose to 450 mg in severe hepatic impairment.

Renal failure is unlikely to affect the pharmacokinetic parameters of alectinib and M4, although no specific PK study has been conducted on this matter. A case report on HD patients receiving alectinib 600 mg twice daily demonstrated that the AUC and $C_{max}$ were similar to those of patients with normal renal function [52]. Alectinib is considered safe for individuals with an eGFR below 30 or who are undergoing HD, even though the package insert does not specifically recommend its use for this population.

**Brigatinib.** During the drug development study, 65% of brigatinib was found to be excreted in feces and 25% was excreted in urine [53]. Brigatinib drug development has involved patients with mild to moderate liver impairment and patients with eGFR greater than 30. The PopPK study demonstrated that there was no significant increase in $AUC_{ss}$ in this population, indicating that the normal dose can be administered as recommended in the package insert [54]. During severe hepatic failure, the unbound AUC of brigatinib was about 37% higher, according to a pharmacokinetic study [55]. Thus, decreasing the dose by around 40% appears to be a reasonable course of action. Adjusting the dosage in cases of hepatic failure is advisable due to brigatinib being primarily metabolised by CYP3A. Therefore, coadministration with moderate or strong CYP3A4 inhibitors should be avoided [56].

In renal failure, unbound brigatinib exposure was approximately 92% higher in patients with severe renal impairment compared with healthy volunteers with normal renal function. This pharmacokinetic study used patients with eGFR range 10–26 [53]. Therefore, reducing the dose to 50% seems reasonable. Initial doses were lower to reduce the hepatotoxicity, which AST and alanine transaminase (ALT) increased can be 34–38% in brigatinib usage, as the package insert stated.

**Ceritinib.** Ceritinib did not have a clinically significant effect on the AUC at steady state ($AUC_{ss}$) in population pharmacokinetic studies that included patients with mild liver impairment and those with an eGFR greater than 30 [57]. Mild renal impairment increases $AUC_{ss}$ by 7%, while moderate renal impairment increases it by 15%. Mild hepatic impairment increases the $AUC_{ss}$ by 3%, but the change is not statistically significant. Ceritinib has been identified as a potent inhibitor of CYP3A and a mild inhibitor of CYP2C9 in a phase I drug-drug interaction study; therefore, caution should be taken when using ceritinib alongside substances metabolized by these enzymes [58]. We included the recommended drug dosage for individuals with mild and moderate renal impairment, as the package insert does not provide dosing instructions for this specific population.

**Crizotinib.** Approximately 53% of unchanged crizotinib is excreted in feces. Approximately 2% of the dose was crizotinib and 5% was O-desalkyl crizotinib lactam are found in the urine [59]. A PopPK study on crizotinib revealed a 9% increase in AUC when comparing patients with total bilirubin levels of 2.1 versus 0.41. However, there were no significant AUC differences observed at different AST levels [60]. A follow-up pharmacokinetic study demonstrated that the AUC and $C_{max}$ of a 250 mg dose twice daily in individuals with normal hepatic function are similar to those of a 250 mg dose twice daily in mild impairment patients, a 200 mg dose twice daily in moderate impairment patients, and a 250 mg dose once daily in severe impairment patients [61]. Therefore, these crizotinib doses can serve as a reference for patients with liver failure.

The AUC of crizotinib was found to be 6% higher in patients with mild renal impairment and 18% higher in patients with moderate renal impairment based on a PopPK study [60]. A PK study was conducted to follow up on patients with severe renal failure. The study showed an 80% increase in the AUC of crizotinib in a single-dose PK study, with subject eGFR ranging from 3.7 to 22.5, and a 54% increase in a multiple-dose PBPK study [62]. Thus, decreasing the dosage of crizotinib by 50% is reasonable in severe renal impairment. There is no data regarding the usage of crizotinib in HD/CAPD.

**Lorlatinib.** Lorlatinib is mainly metabolized by cytochrome P450 (CYP) 3A and UGT1A4, with minimal excretion (<5%) through the kidneys [63]. Currently, there is not enough data available to recommend reducing the dosage for individuals with liver impairment. A PopPK study revealed that individuals with mild impairment have a minimal difference in mean steady-state clearance compared to the normal population, with values of 13.87 L/h and 13.71 L/h, respectively [64]. Thus, individuals with mild liver impairment can receive the standard dosage of lorlatinib.

Despite lorlatinib is only eliminated less than 5% in urine, renal impairment tends to increase clearance in the PopPK study [64]. This finding is supported by a PK study that showed that AUC of lorlatinib increased by 4%, 19%, and 41% in cases of mild, moderate, and severe renal failure, respectively [65]. Reducing the dose to 75% is a reasonable approach in cases of severe renal failure, even though it did not reach statistical significance. This may be due to the small sample size of only five patients, resulting in a wide confidence interval of 141.14% (95%CI = 97.82%, 203.66%).

## BRAF/MEK (v-Raf Murine Sarcoma Viral Oncogene Homolog B/Mitogen-activated Protein Kinase Kinase) inhibitors

One to two percent of lung adenocarcinomas have a BRAF mutation, which is an oncogenic driver. When used to treat BRAF600-mutant metastatic non-small cell lung cancer, dabrafenib, an inhibitor of BRAF, and trametinib, an inhibitor of MEK, had an objective response rate (ORR) of 64% [66]. Vemurafenib, a medication primarily used to treat melanoma, has also demonstrated effectiveness when administered alone to treat NSCLC with BRAF mutations [67]. According to NSCLC guidelines, vemurafenib may be used as an option if a patient is unable to tolerate dabrafenib/trametinib [2].

**Dabrafenib.** 71% of dabrafenib is eliminated in feces, while 23% is eliminated in urine [68]. A PopPK study demonstrated that mild to moderate hepatic impairment had no effect on the clearance divided by bioavailability (Cl/F), an apparent oral clearance parameter. However, this study did not specifically analyze moderate hepatic impairment due to a limited sample size of only three [69]. Therefore, a recommendation for moderate hepatic impairment cannot be drawn from this data. Patients with mild hepatic impairment can safely take the standard dose of dabrafenib, as indicated in the package insert.

A PopPK study found that mild and moderate renal impairment did not significantly affect the Cl/F parameter, with geometric mean percentages of 95% and 92%, respectively [69]. Using the standard dosage of dabrafenib is reasonable. In HD patients, administering dabrafenib at a reduced dose of 50% led to side effects, as demonstrated in a case study [70]. Initiating dabrafenib at the lowest possible dose of 75 mg once daily appears justifiable, as HD did not alter dabrafenib plasma concentration, but thorough evaluation is necessary in this group. Of note, dabrafenib studies only included patients with melanoma, not NSCLC. Additional research is needed to investigate its application in NSCLC.

**Trametinib.** 80% of a radiolabeled dose of trametinib was found in the feces, while less than 20% was found in the urine, with less than 0.1% remaining unchanged [71]. PopPK and PK studies on mild hepatic impairment demonstrated that the standard dose of trametinib had similar pharmacokinetic characteristics and was well-tolerated; thus, it can be administered at the regular dose [72,73]. In cases of moderate and severe hepatic impairment, obtaining samples was challenging due to treatment-related adverse effects. The PK result was not representative of the actual situation due to the small sample size, even though it showed similar $C_{max}$ and AUC properties [73]. No dose-limiting toxicities were observed with 1.5 mg of trametinib per day in patients with moderate hepatic impairment and 1 mg in patients with severe hepatic impairment, in a cohort of at least three patients [73]. This dosage can serve as the starting dose prior to any dosage adjustments.

A PopPK study demonstrated that patients with mild and moderate renal impairment exhibited comparable clearance parameters, Cl/F [72]. Hence, a standard dosage can be utilized. There is no information available on severe renal impairment. A case report demonstrated an adverse reaction to the combination of dabrafenib 75 mg twice daily and trametinib 1 mg once daily. Reducing the dose of trametinib to 0.5 mg once daily resulted in a clinically stable patient with a comparable $C_{max}$ parameter to that of a patient with normal renal function [70]. Thus, commencing trametinib at a minimum dose of 0.5 mg once daily seems reasonable.

**Vemurafenib.** Vemurafenib is a substrate and inducer of CYP3A4, a moderate inhibitor of CYP1A2, and acts as both a substrate and inhibitor of the drug efflux P-glycoprotein (P-gp) and Breast Cancer Resistance Protein (BCRP) transporters [74]. Vemurafenib is primarily eliminated from the body through feces, with less than 1% eliminated through the kidneys. The package insert stated that mild and moderate renal and hepatic impairment had no impact on vemurafenib pharmacokinetics, according to a PopPK study. However, we could not find a publication that supported this statement. The package insert stated that the clearance of vemurafenib was similar in individuals with mild and moderate hepatic impairment, with a total of 58 and 27 subjects, respectively. Clearance in renal impairment was similar among 94 subjects with mild impairment and 11 subjects with moderate impairment.

## NTRK (Neurotrophic Tyrosine Receptor Kinase) inhibitors

The NTRK gene fusion encodes the tropomyosin receptor kinase A (TrKA) protein, which serves as an oncogenic driver in 0.2–0.7% of patients with lung tumors [21]. The NCCN has approved two NTRK inhibitors, larotrectinib and entrectinib, for NSCLC [2]. Two trials demonstrated that the ORR for larotrectinib and entrectinib in solid tumor were 79% and 57%, respectively [75,76].

**Larotrectinib.** Mild liver damage raised Larotrectinib AUC by 1.3 times, while moderate and severe damage increased it by 2 and 3.2 times, respectively. Thus, decreasing the dosage by 50% on the medication label appears suitable. Take additional caution when combining with medications that inhibit organic anion transporting polypeptide (OATP)1A/1B, CYP3A, ATP-binding cassette (ABC)B1 and G2, as they can reduce oral absorption and limit brain accumulation [77]. Avoid using larotrectinib with strong CYP3A4 inhibitors or inducers, as indicated in the package insert. Larotrectinib's AUC and $C_{max}$ increased by 1.5- and 1.3-fold in dialysis compared to normal renal function. No study has been conducted on patients with moderate to severe renal impairment, as indicated in the package insert. However, we were unable to locate any publications that discuss the pharmacokinetic properties mentioned in the package insert statements.

**Entrectinib.** Entrectinib is primarily metabolized and excreted through feces, along with its metabolites. Only 3% of the dose was excreted in urine, indicating minimal renal excretion [78]. The package insert indicated that there were no clinically significant pharmacokinetic parameters in cases of mild to moderate renal impairment and mild hepatic impairment. We were unable to find a specific publication related to this statement. Entrectinib is metabolized primarily by CYP3A4 [79], so precaution is advised when giving entrectinib to patients with liver impairment.

## MET (Mesenchymal-epithelial Transition) inhibitors

MET is a receptor for hepatocyte growth factor (HGF) that plays a role in cellular survival and proliferation. Oncogenic mutations in MET result in suboptimal responses to immunotherapy, regardless of high PD-L1 expression [80]. The ORR of capmatinib was 41% in patients who had undergone prior therapy and 68% in those who had not received previous treatment [81]. The ORR for tepotinib in NSCLC is 57% [82]. Several targeted therapies aimed at MET are currently under development, including glumetinib, Sym015 (a combination of humanized antibodies designed to degrade MET), APL-101, and REGN5093 [21].

**Capmatinib.** Capmatinib is mainly metabolized in the liver, particularly by CYP3A4, and then excreted in feces [83]. A PK study in patients with mild to severe hepatic impairment found no significant difference in AUC and $C_{max}$ of capmatinib compared to healthy subjects [84]. Of note, this pharmacokinetic study used a 200 mg single dose of capmatinib, which is half

of the recommended 400 mg dose. While the standard dose can be administered in cases of liver impairment, it is important to monitor the risk of toxicity associated with capmatinib. The package insert mentioned that mild to moderate renal impairment does not affect pharmacokinetic parameters, but we were unable to locate any publication that supports this claim.

**Tepotinib.** 77.9% of tepotinib was eliminated through faeces, with only 13.6% being excreted through urine [85]. In the PopPK study, individuals with moderate hepatic impairment exhibited a 13% reduction in $AUC_{ss}$ compared to the control group. Furthermore, elevated bilirubin and alkaline phosphatase levels did not cause a clinically significant change in AUC [86]. Therefore, tepotinib can be administered at the standard dosage in cases of mild and moderate hepatic impairment. Additional research is required to evaluate the suitability of tepotinib for individuals with severe hepatic impairment, as the PopPK study did not include this group.

A PopPK study in patients with renal failure found no significant difference in $AUC_{ss}$ between those with eGFR 39.4 to 59 compared to the reference group with eGFR 99.8. The predicted tepotinib $AUC_{ss}$ was measured at 102 (93–112) in this population [86]. This outcome can be used as a reference for determining the appropriate dosage of tepotinib in cases of renal failure. Additional research is required in cases of severe renal failure and dialysis.

### RET (Multiple Endocrine Neoplasia Type 2) inhibitors

RET gene rearrangements occur in 1–2% of NSCLC and are mutually exclusive with mutations in EGFR, ALK, or RAS. NSCLC with RET fusion is linked to an elevated risk of cerebral metastasis [21]. Cabozantinib is the first RET inhibitor approved by the FDA for NSCLC, yet it exhibits an ORR of only 28% [87]. The recent RET inhibitors, selpercatinib and pralsetinib, exhibit higher ORR of 64% and 61%, respectively [88,89].

**Cabozantinib.** 54% of cabozantinib was excreted in feces and 27% in urine [90]. In a PBPK study, the AUC for mild and moderate hepatic impairments increased by 64% and 50%, respectively [91]. In the PK study, these values rose by 81% and 63%, respectively [92]. In a PopPK study, the Cl/F parameter increased by 12% in individuals with mild hepatic impairment, while moderate hepatic impairment did not affect the Cl/F parameter [93]. In addition, the percentage of unbound drug in moderate hepatic impairment was slightly higher at 0.37% compared to 0.24% in mild hepatic impairment and 0.28% in the control group [92]. Based on these results, it is reasonable to reduce the dose in cases of moderate hepatic impairment.

Cabozantinib shows similar pharmacokinetic parameters in individuals with renal impairment and eGFR of 30–89 compared to those with normal kidney function, as indicated by a PK study [92]. Hence, standard dosage can be administered. A case report demonstrated that the AUC in HD was significantly lower compared to the control group (348 vs 1375 ng/mL) [94]. Unfortunately, we cannot make a definitive conclusion about this case report because there is insufficient information regarding the patient's clinical characteristics that may have influenced cabozantinib's pharmacokinetics.

**Selpercatinib.** 69% of selpercatinib was recovered in feces and 24% in urine, with 14% and 12% unchanged, respectively. Selpercatinib AUC increased by 7%, 32%, and 77% in subjects with mild, moderate, and severe impairment, respectively, as stated in the package insert. Therefore, a dose reduction to 80 mg is suitable for severe impairment due to the limited availability of capsules in 40 and 80 mg dosages. No significant differences were observed in selpercatinib pharmacokinetics in cases of renal impairment (eGFR 15–89 mL/min). However, we were unable to locate any publication related to these statements.

**Pralsetinib.** 73% of pralsetinib was eliminated in feces, with 6% being eliminated in urine. A PK study demonstrated that AUC and $C_{max}$ of pralsetinib were similar in moderate and severe hepatic impairments for both total and unbound concentrations [95]. This study observed a 32% decrease in $C_{max}$ in individuals with severe liver impairments, likely attributed to alterations in P-gp activity [96], for which pralsetinib is a substrate. Nonetheless, pralsetinib can be administered at a standard dosage regardless of the severity of liver impairment.

According to the package insert, mild and moderate renal impairment did not affect the exposure of pralsetinib compared to normal renal function. Hence, the standard dosage can be administered to this population. Pralsetinib has not been researched in patients with severe renal impairment. No publication on the pharmacokinetics of praseltinib in patients with renal impairment was identified.

## KRAS (Kirsten Rat Sarcoma Viral) inhibitor

KRAS is the most mutated oncogene in human cancer, occurring in 13% of NSCLC, and it encodes guanosine triphosphatase, which regulates signal transduction. KRAS mutation is frequently linked to resistance against targeted therapies and unfavorable outcomes [21]. The Sotorasib trial, in comparison to docetaxel, indicated that sotorasib can significantly prolong median progression-free survival by 1.1 months and possesses a more advantageous safety profile [97].

**Sotorasib.** Sotorasib is predominantly metabolized by CYP3A4 and is primarily excreted through feces with minimal renal excretion [98]. No significant pharmacokinetic parameters were observed in individuals with mild hepatic impairment or mild to moderate renal impairment, according to the package insert. Therefore, no modification to the dosage is necessary for this group. We were unable to find any publication related to this statement.

## Discussion

NCCN has published clinical practice guidelines for NSCLC and SCLC [2,13]. By combining these guidelines and the result of this study, we can give systemic therapy in the safest way possible for patients with liver impairment or renal impairment. For example, in NSCLC patients with EGFR S768I, L861Q, and/or G719X mutations, the preferred therapy is afatinib or osimertinib [2]. For patients with severe liver impairment, neither afatinib nor osimertinib has been studied, so it is best to take caution when giving these drugs to such patients. In the presence of severe renal impairment, osimertinib can be given in a normal dose, while afatinib dose must be lowered to 30 mg QD in patients undergoing HD and patients with eGFR 15–29 [41,42].

Although the efficacy and safety of targeted therapy have been extensively researched, the pharmacokinetic parameter is more critical as it accurately represents the fate of drugs in the human body. Even minor alterations in factors that affect pharmacokinetic parameters, such as liver/renal function, can have severe consequences, including treatment failure or drug toxicity. Changes in plasma protein levels, CYP enzymes, and treatment adherence can cause variability in the drugs' plasma levels. For instance, in patients with liver metastases, the decreased concentration of albumin can result in higher levels of unbound erlotinib, a pharmacologically active drug, leading to an increased rate of its removal from the body [35,99]. A lower concentration of erlotinib in the bloodstream (<1711 ng/mL) implied a reduced ORR in lung cancer (5% vs 38%; $p = 0.058$) [100]. In addition, elevated alpha-1 acid glycoprotein (AGP) levels (2.3 vs. 1.45 g/L) demonstrated a 15.9% reduction in the clearance of erlotinib [101].

The pathophysiological and biological alterations occurring in liver impairment can also influence the appropriate dosages of targeted therapy administered to the patient. Liver impairment alters the quantity and activity of liver enzymes and transporters, which in turn affects the metabolism of drugs, especially for compounds that necessitate metabolism to generate pharmacologically active metabolites, such as alectinib, a substrate of CYP3A4 [49,102].

Research on pharmacokinetics of targeted therapy in liver impairment commonly uses two primary classifications: Child-Pugh and NCIc. The FDA only explicitly mentions the use of the Child-Pugh criteria, while the European Medicines Agency (EMA) does not provide guidance on categorizing hepatic impairment [103]. The Child-Pugh method is predominantly utilized in targeted therapy pharmacokinetic studies (72%), whereas studies involving cancer patients primarily rely on NCIc (69%) [104]. Targeted therapies that utilize both Child-Pugh and NCIc sometimes have conflicting dosing recommendations because NCIc tends to categorize subjects as less impaired than Child-Pugh [103,104]. Hence, these two classifications should be utilized in exploratory pharmacokinetic analysis, and clinicians should use the classification employed in the pharmacokinetic study. Besides the method used to categorize liver impairment, the cause of liver impairment is an important factor to consider when determining drug dosing. For instance, gefitinib pharmacokinetic parameters were found to vary significantly in cirrhosis but not in liver metastasis [44]. This event is hypothesized to be due to the liver's substantial functional reserve, even when tumor metastasis is present [44,105].

Targeted therapy agents, such as TKIs, have also been associated with hepatotoxicity of different types and levels of severity [106]. The recommendation to adjust dose in patients with liver impairment may differ from one targeted agent to another, as can be seen in Table 1. If liver injury occurs because of the hepatotoxicity of the targeted agents, it is necessary to reduce the dosage. If the injury continues to persist after reducing the dosage, the agent must be discontinued. Guidelines for reducing the dosage and discontinuing the medication were stated in the package insert for each drug.

According to the FDA's guidance for the pharmaceutical industry on liver impairment, conducting a PK study may not be necessary in certain cases. These cases include when a drug is eliminated through the kidneys without any involvement of the liver, when less than 20 percent of the drug is metabolized in the liver and has a wide therapeutic range, or when the drug is in the form of a gas or volatile substance [107]. Due to the high liver metabolism for most oral targeted therapy, PK studies in individuals with liver impairment should be conducted.

In renal impairment, the function of glomeruli and tubules is reduced, resulting in hypoalbuminemia due to loss of protein in the urine and accumulation of uremic toxins in the blood. The uremic toxins affect the drug's distribution and metabolism by inhibiting renal organic anion transporters (OATs) and altering the CYP-mediated metabolism. The clearance of a targeted therapy that is eliminated by the kidney is reduced in renal impairment, prolonging the drug's half-life and increasing the risk of toxicity [102].

We have also discovered a unique result of the lorlatinib study regarding its renal dose. Despite the urinary excretion being less than 5%, lorlatinib AUC significantly rises in cases of renal failure [65]. The cause of this event remains unclear due to the likelihood of renal impairment only affecting the AUC of drugs that undergo extensive metabolism in the liver and have a high hepatic extraction ratio, such as nicardipine [108]. Uremia in renal impairment can disrupt hepatic blood flow, leading to a decrease in the capability of first-pass metabolism [109]. However, lorlatinib has a low extraction ratio of approximately 12% [110], suggesting that the hypothesis may be incorrect.

This lorlatinib result confirms the significance of studying renal impairment, even though renal elimination is low. For drugs that have low urinary excretion, such as lorlatinib, FDA recommend performing a reduced study design. This study design compares patients with severe renal impairment to those with normal renal function. If there are any discrepancies in the AUC, a full PK study incorporating mild and moderate renal impairment should be conducted [103,111]. Hence, it is important to conduct research on renal impairment in all oral targeted therapies for lung disease.

This systematic review also discovered that studies of HD and CAPD primarily consist of case reports and case series, except for one pharmacokinetic study involving three HD subjects [35]. Drug clearance in HD is complicated because of the variability in drug dialyzability and the modified nonrenal drug clearance in dialysis patients [112]. All studies included in this systematic review concluded that oral targeted therapies for lung cancer are not significantly eliminated by dialysis. However, altered nonrenal drug clearance should still be considered an important factor. End-stage renal disease (ESRD) can modify the absorption of oral cancer treatment by affecting paracellular leakage and reducing efflux transporter activity [112]. Uremic toxin can also modify pH and reduce protein binding. Furthermore, changes in albumin levels in kidney disease and during dialysis can affect its protein-binding capacity [112]. Metabolism in ESRD can decrease CYP expression, reducing the body's ability to deactivate the drug [113]. We are encouraging other researchers and drug manufacturers to conduct a PK study in dialysis patients to enhance the utilization in this population.

Evaluating pharmacokinetics in HD or CAPD patients is challenging due to the variability in frailty and comorbidities associated with chronic dialysis. A systematic review of targeted and immune therapies in metastatic renal carcinoma patients undergoing hemodialysis revealed that certain drugs (everolimus and sorafenib) exhibited higher-than-anticipated levels of toxicities in hemodialysis, despite no significant changes in their pharmacokinetic properties [114]. Another retrospective study demonstrated that the use of TKIs in HD patients resulted in a significant increase in adverse events and blood pressure elevation compared to a control group consisting of HD patients who only underwent surgery [115]. Hence, it is essential to closely monitor the administration of oral targeted therapy in patients undergoing dialysis.

A cohort study comparing patients with ESRD to those without ESRD receiving treatment for lung cancer reveals a significant difference in median survival rates: 5.98 months versus 14.13 months, respectively (p = 0.019). The 5.98-month outcome closely matches the median survival of 6.51 months observed in patients with ESRD receiving only supportive care [116]. This outcome could originate from drug toxicity or diminished effectiveness due to sub-therapeutic dosing. A separate study indicated that 44% of patients with ESRD experienced iatrogenic toxicity because of chemotherapy or targeted therapy [117].

According to our systematic review, hepatic and renal impairment should be considered when administering six and seven targeted therapies in lung cancer, respectively. However, ten medications still lacked adequate information on hepatic and renal impairment. Fig 3 provides a summary of these findings. Nevertheless, this recommendation only serves as an initial guiding dose for patients who have renal and hepatic impairment. It is important to regularly evaluate the effectiveness and safety of using oral targeted therapy in patients due to the significant impact of side effects on treatment adherence [118]. Poor compliance with certain targeted agents, such as imatinib, can result in significantly low trough plasma concentration and a reduction in cytogenetic response [99]. Therefore, implementing TDM is a potential strategy for several oral lung cancer therapies. According to a review by Mueller-Schoell et al. (2021), monitoring of drug levels is recommended for gefitinib and trametinib. This review also showed that TDM can also be beneficial for alectinib, crizotinib, erlotinib, and vemurafenib [119]. A retrospective study evaluating TKI drug monitoring revealed significant inter-individual variability among several TKIs, supporting the potential utility of TDM [120].

Another common method for initial drug dosing is the use of pharmacogenetic approaches. Oral targeted therapy is primarily metabolized by CYP3A4 and is also a substrate of P-gp [121,122]. For instance, an elevated expression of P-gp

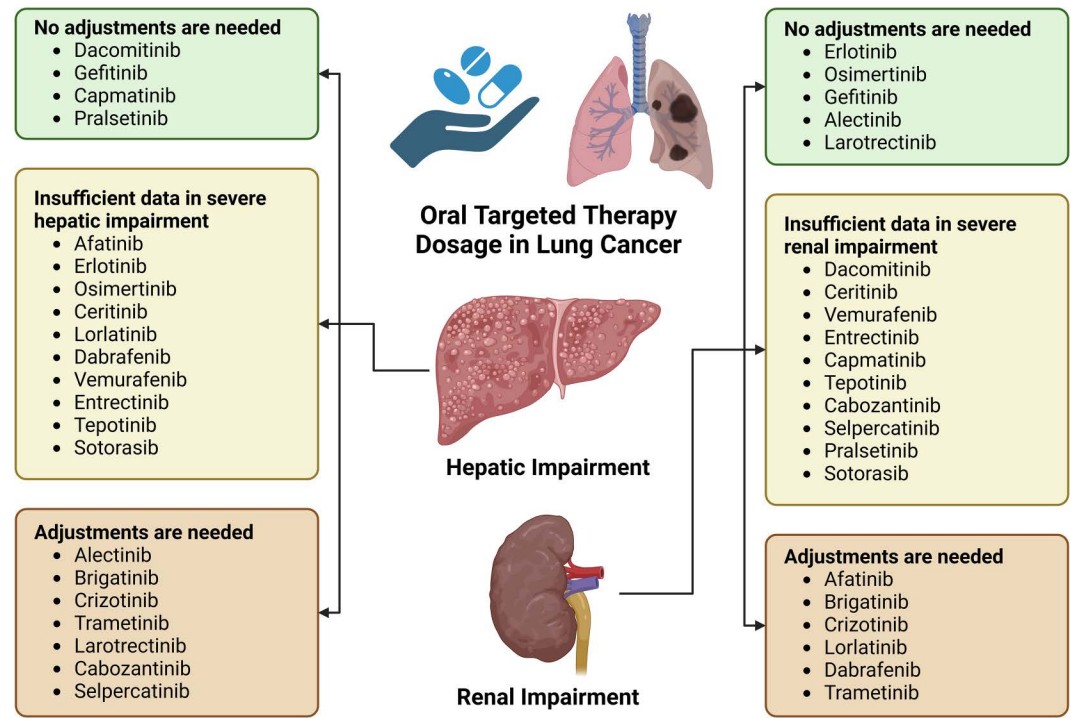

**Fig 3. Summary of findings on dosage of oral targeted therapies for lung cancer in hepatic and renal impairment.** Green box indicates that no adjustment is needed in cases of hepatic or liver impairment. Yellow box indicates the absence of a pharmacokinetic study in individuals with severe hepatic or liver impairment, indicating the need of close monitoring. Red box indicates that oral targeted therapy dosage should be reduced based on either the Child-Pugh score or the estimated glomerular filtration rate.

can reduce oral absorption of larotrectinib [77]. Another example is gefitinib, which is metabolized by CYP2D6. As a result, patients with a poor metabolizer phenotype metabolize gefitinib slower, resulting in a higher rate of hepatotoxicity. In such cases, erlotinib can be used as an alternative [123]. However, pharmacogenetic guidelines for testing ABCB1, CYP3A4, or other metabolizing CYP genes do not exist because of conflicting data regarding the effects of CYP variations on the effectiveness and safety of oral targeted therapy [124]. Pharmacogenetic testing for G6PD may be clinically important when using dabrafenib or trametinib due to their similar sulfonamide structure [125]. However, the latest pharmacogenetic guideline does not recommend routine G6PD testing for these drugs [126].

Thus far, the Clinical Pharmacogenetics Implementation Consortium (CPIC) has not released any guidelines concerning the application of pharmacogenetics in oral targeted therapy for lung cancer. However, while oral targeted therapy is a relatively novel pharmacological intervention, additional study may be beneficial for the development of guidelines and its integration into real-world settings. Conversely, the application of TDM in oral targeted therapy should be enhanced, given the large interindividual variability (IIV) in exposure at conventional doses, which predominantly ranges from 19% to 100% and can reach up to a 16-fold difference for gefitinib. Moreover, evaluating efficacy is a challenge, as OS and PFS assessments need long periods for examination; thus, TDM could serve as a viable proxy for attaining beneficial results [119].

Regarding drug interaction, concurrent use of CYP3A4 and P-gp inhibitors or inducers can significantly affect some oral targeted therapy pharmacokinetic parameters [127]. In cases of chronic renal failure, the expression of P-gp is notably elevated compared to individuals with normal renal function [128]. Conversely, acute hepatic failure reduces the level of intestinal P-gp expression [129]. Certain oral targeted therapies also function as inhibitors for specific enzymes in the CYP450 family. For example, gefitinib inhibits CYP2C19 and 2D6, vemurafenib inhibits CYP1A2 and 2D6, and crizotinib inhibits CYP3A4 [127]. Therefore, it is important to use caution when using these medications in conjunction with other specific CYP450 substrates. Certain oral targeted therapies exhibit enhanced absorption when exposed to an acidic stomach environment. As can be seen in a nationwide cohort study, the simultaneous use of a proton pump inhibitor and gefitinib will significantly decrease OS of lung cancer patients [130]. These factors should be taken into consideration when tailoring a dosing regimen for a patient.

Some studies in this systematic review focused on populations other than those with lung cancer, as certain drugs are primarily used in different types of cancer. A pharmacokinetic study on cabozantinib revealed that medullary thyroid carcinoma led to a 93% increase in apparent clearance compared to other types of cancer [131]. Hypocalcemia in thyroid carcinoma is hypothesized to impact the clearance of cabozantinib. Another PopPK study on Avelumab found that metastatic Merkel cell carcinoma and squamous cell carcinoma had reduced clearance rates of 32% and 25%, respectively, compared to other types of cancer [132]. Some cancers may induce pharmacokinetic alterations that impact the metabolism and elimination of targeted therapy, so findings from studies in different cancer types should be interpreted with caution.

This study has several limitations. First, our focus was on studies that included pharmacokinetic analysis. Consequently, studies that solely investigated clinical effectiveness and side effects were deliberately excluded. Studies on the use of oral targeted therapy without pharmacokinetic analysis can be applicable in patients, albeit with less reliable evidence compared to studies that include pharmacokinetic analysis. Second, we encountered a lack of available publications related to package insert claims for some drugs. Third, we incorporated certain studies that were not conducted specifically in the lung cancer population, thereby augmenting the potential for bias in the results related to lung cancer. Fourth, certain studies exhibit a small number size, particularly in their dependence on small case series for dialysis data. Fifth, there may be a publication bias favoring positive pharmacokinetic study outcomes, as most of the included studies were conducted by researchers affiliated with the medication manufacturer. Finally, variability may arise from the utilization of distinct liver impairment classifications (Child-Pugh vs. NCIc).

Manufacturers can engage with consortia to conduct research that addresses a lack of publications regarding targeted therapy in lung cancer for patients with renal and hepatic impairment. Currently, there are 15 consortia engaged in early-phase oncology clinical trials [133]. This collaboration may address the difficulties of enrolling appropriate patients and

initiating the process. Collaboration may also be undertaken for the execution of case reports or case series pharmacokinetic studies. A case report may serve as a foundation for subsequent clinical pharmacokinetic studies concerning hepatic and renal dysfunction. Moreover, employing alternative methodologies through PopPK studies, if patients with hepatic impairment are not excluded from phase 2 and 3 trials and sufficient pharmacokinetic data is gathered, represents a viable way to enhance publication in this area [103]. Finally, the application of PBPK modeling in drug development and dosing is recommended to improve research and publishing of drug dosage in cases of hepatic or renal impairment.

According to FDA guidance, PBPK can serve as an alternative for the preliminary characterization of drug dosing regimens. Subsequently, clinical trials may be conducted to validate the outcomes of PBPK [111]. Currently, the application of PBPK in organ impairment remains small, constituting 8% of total PBPK utilization. PBPK modeling has been utilized to guide dosing in drug labels for at least 109 new drugs as of 2023, 54 of which correspond to the oncology field [134]. The application of PBPK modeling in oncology is promising, as a systematic review indicated that the absolute average folding errors of AUC between predicted values from PBPK and observed values from clinical trials of organ impairment were within 1.15-fold of each other, which is a notably favorable result [135].

To summarize, the dosage of some oral targeted therapy for lung cancer should be decreased in cases of hepatic or renal impairment. This systematic review has generated multiple new recommendations and revised existing recommendations regarding the adjustment of oral targeted therapy initial doses in the presence of hepatic and renal impairment. Additional factors that need to be considered include hypoalbuminemia, hepatotoxicity, and drug interactions. It is highly recommended to conduct additional pharmacokinetic research on the usage of this drug in populations that have not been studied before.

## Supporting information

**S1 File. Preferred Reporting Items for Systematic Reviews and Meta-Analyses (PRISMA) checklist.**
(DOCX)

**S2 File. Supplementary materials.**
(DOCX)

## Author contributions

**Conceptualization:** Harri Hardi, Hana Khairina Putri Faisal, Vivian Soetikno.

**Data curation:** Harri Hardi, Zahra Fitrianti, Karen Elliora Utama, Ananda Pipphali Vidya, Nurul Gusti Khatimah.

**Formal analysis:** Harri Hardi, Zahra Fitrianti, Karen Elliora Utama, Ananda Pipphali Vidya, Hana Khairina Putri Faisal, Vivian Soetikno.

**Funding acquisition:** Vivian Soetikno.

**Investigation:** Harri Hardi, Zahra Fitrianti, Karen Elliora Utama, Ananda Pipphali Vidya, Nurul Gusti Khatimah.

**Methodology:** Harri Hardi, Hana Khairina Putri Faisal, Vivian Soetikno.

**Project administration:** Harri Hardi.

**Resources:** Harri Hardi, Vivian Soetikno.

**Software:** Harri Hardi.

**Supervision:** Hana Khairina Putri Faisal, Vivian Soetikno.

**Validation:** Hana Khairina Putri Faisal, Vivian Soetikno.

**Visualization:** Harri Hardi.

**Writing – original draft:** Harri Hardi, Zahra Fitrianti, Kevin Aristyo.

**Writing – review & editing:** Hana Khairina Putri Faisal, Vivian Soetikno.

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
