## [Decision Letter · Decision Letter 0]

Dear Dr. Soetikno,

Thank you for submitting your manuscript to PLOS ONE. After careful consideration, we feel that it has merit but does not fully meet PLOS ONE’s publication criteria as it currently stands. Therefore, we invite you to submit a revised version of the manuscript that addresses the points raised during the review process.

We look forward to receiving your revised manuscript.

Kind regards,

Babatomiwa Kikiowo

Academic Editor

PLOS ONE

Additional Editor Comments (if provided):

Reviewers' comments:

Reviewer's Responses to Questions

**Comments to the Author**

1. Is the manuscript technically sound, and do the data support the conclusions?

Reviewer #1: Yes

Reviewer #2: Yes

Reviewer #3: Yes

2. Has the statistical analysis been performed appropriately and rigorously?

Reviewer #1: Yes

Reviewer #2: N/A

Reviewer #3: N/A

3. Have the authors made all data underlying the findings in their manuscript fully available?

Reviewer #1: No

Reviewer #2: Yes

Reviewer #3: Yes

4. Is the manuscript presented in an intelligible fashion and written in standard English?

Reviewer #1: Yes

Reviewer #2: Yes

Reviewer #3: Yes

Reviewer #1: There are minor inconsistencies in the body as well as some experiments or conclusions not explained. Great work overall, but kindly ensure you explain some of the results in a more details fashion.

For example, 'A PK study in HD patients showed no significant difference in AUC and Cmax'. Why do you think this happened? Would like a more detailed discussion on this.

Work is also scientifically sound but can be made better when the explanations/discussions are clearly detailed out.

Lastly, a minor observation is some abbreviations were not defined in the body of work. Fixing this would be great.

Reviewer #2: Lung cancer, the foremost cause of cancer-related deaths globally, presents significant management challenges, especially in its advanced stage. This systematic review focuses on the dosage adjustments of targeted therapies for patients with concurrent renal and hepatic impairments, a frequent complication in cancer patients. By examining the package insert recommendations against the backdrop of available pharmacokinetic studies, the authors provide a comprehensive evaluation of the current practices. However, my only wish is that the authors had included a figure to summarize the work, which would have made it more appealing and accessible to the readers

Reviewer #3: The manuscript is well-prepared. However, I recommend the following revisions:

Expand the Methods section to provide more detailed information.

Ensure that the PRISMA guidelines are highlighted and adhered to within the Methods section.

Including the chemical structure of each drug in the Results section could enhance readers' understanding.

**Do you want your identity to be public for this peer review?** For information about this choice, including consent withdrawal, please see our Privacy Policy

Reviewer #1: No

Reviewer #2: No

Reviewer #3: No

---

## [Author Response · Author response to Decision Letter 1]

28 Aug 2024

Comments to the Author

Reviewer #1: There are minor inconsistencies in the body as well as some experiments or conclusions not explained. Great work overall, but kindly ensure you explain some of the results in a more details fashion.

For example, 'A PK study in HD patients showed no significant difference in AUC and Cmax'. Why do you think this happened? Would like a more detailed discussion on this.

This paragraph already contains an elucidation of this phenomenon. Initially, it appears that renal failure does not affect the elimination of erlotinib. Therefore, there is no change in the elimination of erlotinib in end stage renal disease patients, who are typically undergoing hemodialysis. Nevertheless, a study conducted on hemodialysis patients revealed a tendency towards a decrease in the area under the curve (AUC). This could be attributed to hypoalbuminemia, a condition commonly observed in patients undergoing hemodialysis, which leads to a decrease in the overall concentration of erlotinib in the body.

The explanation of hypoalbuminemia condition related to drug elimination also has been explained in discussion.

Work is also scientifically sound but can be made better when the explanations/discussions are clearly detailed out.

We have incorporated three key points into the discussion.

1. The FDA guidance on conducting drug studies for liver impairment and recommendations to manufacturers.

2. The FDA guidance on conducting drug studies for renal impairment and recommendations to manufacturers.

3. Factors to consider in patients with end-stage renal disease undergoing hemodialysis (HD) or continuous ambulatory peritoneal dialysis (CAPD)

Presumably, this addition has ability to enhance the comprehensiveness of our study. If there are any discussions that you believe should be included, please inform us.

Lastly, a minor observation is some abbreviations were not defined in the body of work. Fixing this would be great.

We have added HD, CAPD, Cl/F, ALP, CYP, p-gp, BCRP, AUCss and PRISMA abbreviation in the manuscript. We have assessed all manuscript and have put all abbreviation definition in the first of this abbreviation exist

Reviewer #2: Lung cancer, the foremost cause of cancer-related deaths globally, presents significant management challenges, especially in its advanced stage. This systematic review focuses on the dosage adjustments of targeted therapies for patients with concurrent renal and hepatic impairments, a frequent complication in cancer patients. By examining the package insert recommendations against the backdrop of available pharmacokinetic studies, the authors provide a comprehensive evaluation of the current practices. However, my only wish is that the authors had included a figure to summarize the work, which would have made it more appealing and accessible to the readers

I greatly appreciate your review. Figure 2 and its corresponding explanation have been included in the manuscript, as shown below. In addition, we have incorporated the utilization of Biorender as our software for creating this Figure in the method section.

Reviewer #3: The manuscript is well-prepared. However, I recommend the following revisions:

Expand the Methods section to provide more detailed information.

We greatly appreciate your review. We have reviewed the Methods section and believe that it is adequately detailed to allow reproducibility by other researchers. Additionally, we have included information about the use of resources such as PubChem and BioRender in our systematic review, which can be found in the manuscript. We also have included more comprehensive inclusion and exclusion criteria and why we used these criteria. If there are any specific aspects of the Methods section that you feel require further elaboration, please let us know, and we will gladly address them.

Ensure that the PRISMA guidelines are highlighted and adhered to within the Methods section.

After conducting a thorough review of the PRISMA guideline, we have ensured that all applicable items have been completed in our systematic review. The only exception is the statistical analysis, which can only be performed in a meta-analysis. The comprehensive information regarding our PRISMA guideline adherence can be found in File S1.

Including the chemical structure of each drug in the Results section could enhance readers' understanding.

We have added the chemical structure and PubChem CID of each drug in Table 1.

---

## [Decision Letter · Decision Letter 1]

Dear Dr. Soetikno,

Thank you for submitting your manuscript to PLOS ONE. After careful consideration, we feel that it has merit but does not fully meet PLOS ONE’s publication criteria as it currently stands. Therefore, we invite you to submit a revised version of the manuscript that addresses the points raised during the review process.

We look forward to receiving your revised manuscript.

Kind regards,

Babatomiwa Kikiowo

Academic Editor

PLOS ONE

Journal Requirements:

Reviewers' comments:

Reviewer's Responses to Questions

**Comments to the Author**

Reviewer #1: All comments have been addressed

Reviewer #3: All comments have been addressed

Reviewer #4: All comments have been addressed

Reviewer #5: All comments have been addressed

2. Is the manuscript technically sound, and do the data support the conclusions?

Reviewer #1: Yes

Reviewer #3: Yes

Reviewer #4: Yes

Reviewer #5: Yes

3. Has the statistical analysis been performed appropriately and rigorously?

Reviewer #1: Yes

Reviewer #3: I Don't Know

Reviewer #4: N/A

Reviewer #5: No

4. Have the authors made all data underlying the findings in their manuscript fully available?

Reviewer #1: Yes

Reviewer #3: Yes

Reviewer #4: Yes

Reviewer #5: Yes

5. Is the manuscript presented in an intelligible fashion and written in standard English?

Reviewer #1: Yes

Reviewer #3: Yes

Reviewer #4: Yes

Reviewer #5: Yes

Reviewer #1: Looks good now. The major points were raised and the tables also looked good. The work is also interesting and worthy of being showcased.

Reviewer #3: No comments, everything looks good.

Reviewer #4: The authors presented their findings on "Tailoring Oral Targeted Therapies Dosage in Lung Cancer: A Systematic Review of Pharmacokinetics Studies on Renal and Hepatic Impairment." While the work is scientifically sound, there are opportunities for enhancement.

To improve the manuscript, I recommend that the authors provide more detailed explanations and discussions throughout the text. Clarifying the implications of the findings, particularly regarding how renal and hepatic impairments influence drug dosing, will enrich the overall narrative. Additionally, including more context on the relevance of pharmacokinetics in tailoring therapies for lung cancer patients would strengthen the manuscript's impact. Overall, a more thorough discussion will make the work more accessible and informative for readers.

Reviewer #5: There are minor inconsistencies in your statistical analysis. Great work overall, but kindly ensure to include figures that explain your statistical analysis.

**Do you want your identity to be public for this peer review?** For information about this choice, including consent withdrawal, please see our Privacy Policy

Reviewer #1: No

Reviewer #3: No

Reviewer #4: No

Reviewer #5: No

---

## [Author Response · Author response to Decision Letter 2]

3 Dec 2024

Comments to the Author

Reviewer #4: The authors presented their findings on "Tailoring Oral Targeted Therapies Dosage in Lung Cancer: A Systematic Review of Pharmacokinetics Studies on Renal and Hepatic Impairment." While the work is scientifically sound, there are opportunities for enhancement.

To improve the manuscript, I recommend that the authors provide more detailed explanations and discussions throughout the text. Clarifying the implications of the findings, particularly regarding how renal and hepatic impairments influence drug dosing, will enrich the overall narrative.

I greatly appreciate your insightful review. We have reviewed our manuscript and revised several typographical errors regarding the influence of renal impairment on drug dosing. Hopefully, it increases the understanding of drug dosing in renal impairment.

We have added an explanation regarding how liver impairment can influence drug dosing, with the aim of enhancing readability and comprehension.

Additionally, including more context on the relevance of pharmacokinetics in tailoring therapies for lung cancer patients would strengthen the manuscript's impact. Overall, a more thorough discussion will make the work more accessible and informative for readers.

We have included information regarding the necessity of personalizing targeted therapy dosages in lung cancer, exemplified by erlotinib, one of the most prevalent medications for this condition. Low blood levels of erlotinib markedly reduced the objective response rate (ORR), whereas excessive dosages substantially heightened its toxicity.

Consequently, certain targeted therapies have been recommended for regular monitoring through therapeutic drug monitoring (TDM) to mitigate side effects and enhance efficacy, as elucidated in the subsequent paragraph.

We have also included a discussion on the significance of drug dosing in end-stage renal disease. It is hoped that this will enhance the significance of our systematic review in the future development of targeted therapy research.

Reviewer #5: There are minor inconsistencies in your statistical analysis. Great work overall, but kindly ensure to include figures that explain your statistical analysis.

We sincerely appreciate your feedback on our manuscript. We acknowledge that our systematic review in pharmacokinetic research methods may be unfamiliar. Thus, we present a graph to illustrate research methodologies, as depicted in Figure 1. This manuscript is a systematic review, not a meta-analysis; therefore, we do not conduct statistical analysis. Consequently, we are unable to provide figures that elucidate our statistical analysis.

We sincerely appreciate your feedback on our manuscript. We acknowledge that our systematic review in pharmacokinetic research methods may be unfamiliar. Thus, we present a graph to illustrate research methodologies, as depicted in Figure 1. This manuscript is a systematic review, not a meta-analysis; therefore, we do not conduct statistical analysis. Consequently, we are unable to provide figures that elucidate our statistical analysis.

---

## [Decision Letter · Decision Letter 2]

Dear Dr. Soetikno,

Thank you for submitting your manuscript to PLOS ONE. After careful consideration, we feel that it has merit but does not fully meet PLOS ONE’s publication criteria as it currently stands. Therefore, we invite you to submit a revised version of the manuscript that addresses the points raised during the review process.

We look forward to receiving your revised manuscript.

Kind regards,

Zheng Yuan

Academic Editor

PLOS ONE

Journal Requirements:

Comments from PLOS Editorial Office: As your systematic review aims to inform clinical decision making, please include a certainty of evidence assessment (e.g.: GRADE) in your revised manuscript.

Reviewers' comments:

Reviewer's Responses to Questions

**Comments to the Author**

Reviewer #1: All comments have been addressed

Reviewer #5: All comments have been addressed

Reviewer #6: (No Response)

Reviewer #7: (No Response)

2. Is the manuscript technically sound, and do the data support the conclusions?

Reviewer #1: Yes

Reviewer #5: Yes

Reviewer #6: Yes

Reviewer #7: Yes

3. Has the statistical analysis been performed appropriately and rigorously?

Reviewer #1: Yes

Reviewer #5: Yes

Reviewer #6: Yes

Reviewer #7: Yes

4. Have the authors made all data underlying the findings in their manuscript fully available?

Reviewer #1: Yes

Reviewer #5: Yes

Reviewer #6: Yes

Reviewer #7: Yes

5. Is the manuscript presented in an intelligible fashion and written in standard English?

Reviewer #1: Yes

Reviewer #5: Yes

Reviewer #6: Yes

Reviewer #7: Yes

Reviewer #1: my concerns have been duly worked on. I recommend for publication. In addition to other reviewers comments, I believe the authors duly made their writing and communication much clearer, wit an added emphasis on proper data visualization.

Reviewer #5: The review is thorough, covering a broad range of oral targeted therapies for lung cancer in patients with hepatic and renal impairment.

Table 1, which talks about summary of dose adjustments, is critical but lacks an interpretation section.

Figure 3 could be referenced more explicitly in the discussion. (Instead of saying "Key findings are summarized in Fig. 3," specifying which insights the figure conveys makes more sense in my opinion.)

Reviewer #6: Great work overall, with succinct analysis. I have a few comments that would further enhance the flow and impact of the article.

1. In your work, you effectively categorized targeted therapies under various specific considerations, such as EGFR, ALK, BRAF/MEK, etc., and provided key conclusions for each therapy within these sections. To enhance the reader's comprehension, it would be immensely beneficial to include a brief explanation at the beginning of each section about these considerations. For example, a concise description of EGFR inhibitors (what they are and why they are important in targeted therapy) would provide essential context before delving into the analysis of specific therapies.

2. For Table 1, consider adding an additional column to include details on the inhibitory considerations (e.g., EGFR, ALK, etc.) for each therapy. While I understand that space limitations might make this challenging, it might be informative. This is a suggestion not a strict recommendation.

3. While one of the key challenges resulting to limited studies in the pharmacokinetic analysis of the drugs due to lack of available data from manufacturers was highlighted, your discussion would benefit from mentioning more key challenges for the limited studies. Additionally, providing recommendations for researchers and manufacturers on overcoming these challenges could enhance the practical value of your discussion. This would elevate your work beyond the already well-articulated dosage recommendations, giving it broader applicability and impact.

Reviewer #7: 1. Merits

Comprehensive Scope

The manuscript systematically surveys pharmacokinetic (PK) studies for 21 oral targeted therapies in lung cancer patients with renal and hepatic impairment, spanning PBPK, PopPK, PK, and case‐series designs. This breadth provides clinicians with a one‐stop reference for dose‐adjustment guidance.

Rigorous Methodology

Adherence to PRISMA guidelines and PROSPERO registration (CRD42024518123) underscore the authors’ commitment to transparency and reproducibility. Dual‐reviewer screening, standardized data extraction, and the use of established risk‐of‐bias tools strengthen the review’s credibility.

Novel Contribution

To our knowledge, this is the first systematic review focusing exclusively on PK–based dose recommendations for targeted lung cancer therapies in patients with organ impairment. The 16 new and 8 updated dosing recommendations add tangible value beyond existing package-insert guidance.

Clinical Relevance

By juxtaposing label recommendations against empirical PK data—including in hemodialysis and peritoneal dialysis—this work has immediate applicability in oncology practice, aiding in precision dosing to maximize efficacy and safety.

2. Aspects for Improvement

Search Strategy Transparency

Detail the full search strings in the main Methods (or move from Table S1 into a Supplement with explicit Boolean operators and date ranges).

Clarify database coverage (e.g., specify date of last search for Web of Science, Scopus, PubMed).

Risk‐of‐Bias Synthesis

Summarize key findings of the bias assessment in the Results rather than relegating details to S4–S6 Tables. A brief paragraph quantifying how many studies were at low vs. moderate vs. high risk will help readers gauge confidence in the recommendations.

Results Presentation

Unify and simplify dosing tables. Currently, Table 1 and S7 appear to overlap. A consolidated main-text table with columns “Drug | Impairment Category | ΔAUC/Cmax | Dose Recommendation” would increase readability.

Include a PRISMA flow diagram as Figure 2 in the main text rather than only referencing it.

Discussion Depth & Limitations

Expand on clinical decision-making. For example, discuss how TDM or pharmacogenetic testing might integrate with these dosing recommendations in real‐world settings.

Explicitly state limitations. The manuscript currently truncates before a standalone “Limitations” subsection. Address heterogeneity in study designs, missing data for certain eGFR or Child–Pugh classes, and language/publication bias.

Manuscript Organization & Clarity

Remove embedded “Response to Reviewers” text from the manuscript draft. That content belongs in your cover letter or tracked-changes response, not the submitted article itself.

Proofread for redundancy and English usage. A few sections (e.g., Reviewer #5 responses) are duplicated verbatim. Consolidate and streamline to avoid reader confusion.

Figures & Visual Summary

Add a summary schematic (e.g., a heatmap or color-coded chart) showing which drugs require dose reduction by impairment severity.

Ensure all figures (including the Biorender diagram) are high resolution and clearly labelled.

**Do you want your identity to be public for this peer review?** For information about this choice, including consent withdrawal, please see our Privacy Policy

Reviewer #1: No

Reviewer #5: **Yes: ** Stephen Adepoju

Reviewer #6: No

Reviewer #7: **Yes: ** Xiaoyi Zhang, MD.

---

## [Author Response · Author response to Decision Letter 3]

26 May 2025

I greatly appreciate for your time and consideration to this manuscript. All of reviewer concern has been addressed in attachment "response to reviewer"

---

## [Decision Letter · Decision Letter 3]

Dear Dr. Soetikno,

Thank you for submitting your manuscript to PLOS ONE. After careful consideration, we feel that it has merit but does not fully meet PLOS ONE’s publication criteria as it currently stands. Therefore, we invite you to submit a revised version of the manuscript that addresses the points raised during the review process.

We look forward to receiving your revised manuscript.

Kind regards,

Zheng Yuan

Academic Editor

PLOS ONE

**Journal office comment to authors: **

Thank you for submitting your revised manuscript. However, we note that our previous request "As your systematic review aims to inform clinical decision making, please include a certainty of evidence assessment (e.g.: GRADE) in your revised manuscript." has been overlooked. 

We do note that you included a quality assessment, but  a certainty of evidence assessment is also required for your study. Quality assessment focuses on evaluating the internal validity of individual studies included in a review, looking for biases and limitations in their design, execution, and reporting. Certainty of evidence, on the other hand, refers to the overall confidence in the findings of the review, considering the quality of the included studies, but also the consistency of results, and other factors that might impact the reliability of the findings. 

Given that your systematic review aims to inform clinical decision making, please also include a certainty of evidence assessment (e.g.: GRADE) in your revised manuscript.

Reviewers' comments:

Reviewer's Responses to Questions

**Comments to the Author**

Reviewer #7: (No Response)

2. Is the manuscript technically sound, and do the data support the conclusions?

Reviewer #7: Partly

3. Has the statistical analysis been performed appropriately and rigorously?

Reviewer #7: Yes

4. Have the authors made all data underlying the findings in their manuscript fully available?

Reviewer #7: Yes

5. Is the manuscript presented in an intelligible fashion and written in standard English?

Reviewer #7: Yes

Reviewer #7: Merits

1. Timely and clinically relevant topic

Addresses a growing need to optimize dosing of oral targeted agents in patients with liver and kidney dysfunction, a population often excluded from pivotal trials.

Provides practical guidance that could directly inform oncologists and clinical pharmacologists.

2. Comprehensive methodology

Protocol registered in PROSPERO (CRD42024518123) and adherence to PRISMA standards enhance transparency.

Multi‐database search (PubMed, Scopus, Web of Science) with no language restrictions ensures broad coverage.

3. Risk‐of‐bias assessment

Use of established tools (Murad’s RoB for case reports, NOS for PK studies, PROBAST for PopPK/PBPK) demonstrates rigor in evaluating study quality.

4. Structured presentation of findings

Results organized by drug class, with clear tables summarizing AUC/Cmax changes and recommended dosage modifications.

Inclusion of a visual schematic (Fig. 3) and BioRender diagrams aids reader comprehension.

5. Integration of package‐insert and literature data

Highlights discrepancies between label recommendations and published PK studies, and proposes updated dosing advice where evidence allows.

Aspects for Improvement

1. Search strategy transparency

The main text refers to “Comprehensive keyword usage” in Table S1 but does not detail full search strings.

Suggestion: Move critical elements of the search strategy (e.g., exact terms, date ranges) into the main Methods or an accessible appendix.

2. Handling of heterogeneity and narrative synthesis

While heterogeneity precludes meta‐analysis, the narrative discussion often lists findings without synthesizing overarching patterns.

Suggestion: Introduce summary tables or decision algorithms grouping drugs by degree of impairment (e.g., no adjustment vs. dose reduction vs. avoid) to guide clinicians.

3. Limitations section

A concise, dedicated Limitations paragraph is missing. Key issues include: publication bias toward positive PK results, reliance on small case series for dialysis data, and variability in impairment classification (Child–Pugh vs. NCIc).

Suggestion: Explicitly acknowledge these and discuss their impact on generalizability.

4. Evidence gaps and unsupported statements

Several package‐insert claims (e.g., for vemurafenib, entrectinib, selpercatinib) lack citable PK publications.

Suggestion: Either provide citations or rephrase these as “manufacturer‐reported” and flag the need for independent confirmation.

5. Clarity and conciseness of Discussion

The Discussion is lengthy and occasionally repetitive, particularly in describing pathophysiology.

Suggestion: Streamline by focusing on the most clinically actionable points and moving detailed mechanistic explanations to a separate subsection or supplemental material.

6. Formatting and adherence to journal guidelines

Financial disclosure and ethics statements appear as placeholders in the submission form but are not integrated into the manuscript text.

Suggestion: Ensure that all PLOS ONE submission requirements (funding sources, competing interests, data availability) are clearly stated in the manuscript.

7. Consistency in dosing recommendations

In some sections, you propose new recommendations that differ subtly from package insert guidance without clear rationale.

Suggestion: For each proposed change, briefly summarize the magnitude of PK alterations (e.g., “AUC↑50% in CP C, therefore reduce dose by 50%”) to justify the recommendation.

**Do you want your identity to be public for this peer review?** For information about this choice, including consent withdrawal, please see our Privacy Policy

Reviewer #7: **Yes: ** Xiaoyi Zhang, M.D.

---

## [Author Response · Author response to Decision Letter 4]

30 Jun 2025

I greatly appreciated the time and your consideration for this manuscript. All of the concern has been addressed in the attachment. We hope that this revision is adequately PLoS one publication criteria

---

## [Decision Letter · Decision Letter 4]

Tailoring Oral Targeted Therapies Dosage in Lung Cancer: A Systematic Review of Pharmacokinetics Studies on Renal and Hepatic Impairment

PONE-D-24-21717R4

Dear Dr. Soetikno,

We’re pleased to inform you that your manuscript has been judged scientifically suitable for publication and will be formally accepted for publication once it meets all outstanding technical requirements.

Kind regards,

Zheng Yuan

Academic Editor

PLOS ONE

Additional Editor Comments (optional):

Reviewers' comments:

Reviewer's Responses to Questions

**Comments to the Author**

Reviewer #7: All comments have been addressed

2. Is the manuscript technically sound, and do the data support the conclusions?

Reviewer #7: Yes

3. Has the statistical analysis been performed appropriately and rigorously?

Reviewer #7: Yes

4. Have the authors made all data underlying the findings in their manuscript fully available?

Reviewer #7: Yes

5. Is the manuscript presented in an intelligible fashion and written in standard English?

Reviewer #7: Yes

Reviewer #7: (No Response)

**Do you want your identity to be public for this peer review?** For information about this choice, including consent withdrawal, please see our Privacy Policy

Reviewer #7: **Yes: ** Xiaoyi Zhang

---

## [Editor Report · Acceptance letter]

PONE-D-24-21717R4

PLOS ONE

Dear Dr. Soetikno,

I'm pleased to inform you that your manuscript has been deemed suitable for publication in PLOS ONE. Congratulations! Your manuscript is now being handed over to our production team.

Kind regards,

on behalf of

Dr. Zheng Yuan

Academic Editor

PLOS ONE